# Percutaneous Biopsy Under Deep Intravenous or Oral Conscious Sedation: Which Is the Best Option for Pediatric Renal Transplant Recipients?

**DOI:** 10.3390/jcm14207361

**Published:** 2025-10-17

**Authors:** Nicola Bertazza Partigiani, Anna Zanin, Beatrice Martini, Benedetta Antoniello, Susanna Negrisolo, Maria Sangermano, Franca Benini, Elisa Benetti

**Affiliations:** 1Pediatric Nephrology Unit, Department of Women’s and Children’s Health, Padua University Hospital, 35128 Padua, Italy; nicola.bertazzapartigiani@aopd.veneto.it (N.B.P.); maria.sangermano@aopd.veneto.it (M.S.); elisa.benetti@aopd.veneto.it (E.B.); 2Laboratory of Immunopathology and Molecular Biology of the Kidney, Pediatric Research Institute “IRP Città della Speranza”, Department of Women’s and Children’s Health, Padua University Hospital, 35128 Padua, Italysusanna.negrisolo@unipd.it (S.N.); 3Palliative Care and Pain Service, Department of Women’s and Children’s Health, Padua University Hospital, 35128 Padua, Italy

**Keywords:** percutaneous kidney biopsy, kidney transplantation, sedation, children

## Abstract

**Background:** Renal allograft biopsy is essential in the follow-up of pediatric kidney transplant recipients, but the optimal sedation strategy remains uncertain. **Methods:** We retrospectively reviewed 711 ultrasound-guided biopsies in 251 children and adolescents (2009–2024), comparing oral conscious sedation with midazolam to deep intravenous (IV) sedation with propofol, midazolam, and ketamine. Outcomes included tissue yield, diagnostic success, complications, and cost-effectiveness. **Results:** IV sedation was used in 77.1% of procedures and was associated with longer cortical cores (median 1.8 vs. 1.5 cm, *p* < 0.001) and more glomeruli (16 vs. 8, *p* < 0.001), improving tissue yield and consequently increasing diagnostic success from 75% to 88.5% (*p* < 0.001; OR 2.6). Biopsy-related complications occurred in 12.9% of cases, with no difference between groups. Sedation-related complications, all mild or moderate, occurred only with IV sedation (4.9%). The improved tissue yield reduced the cost per successful diagnosis (EUR 1243 vs. EUR 1467), making IV sedation the dominant strategy. **Conclusions:** IV sedation enhances the diagnostic quality and cost-effectiveness of pediatric kidney allograft biopsies without increasing overall risk, though prospective studies should also assess patient anxiety and comfort.

## 1. Introduction

Percutaneous kidney biopsy is a cornerstone in the management of pediatric kidney transplant recipients, as it provides histological information essential for diagnosing rejection, monitoring disease recurrence, and tailoring immunosuppressive therapy, thereby ensuring long-term graft survival [1,2]. In children, these procedures are particularly relevant, since early recognition of graft pathology may allow prompt treatment and optimize outcomes in a population with lifelong risk of graft failure. Despite being considered minimally invasive, kidney transplant biopsies in children are technically and emotionally challenging because of younger age, anxiety, and limited ability to remain still [3]. Adequate sedation is therefore fundamental to guarantee immobility, safety, and comfort, as well as to optimize tissue yield and minimize procedure-related complications. A wide range of approaches is used, from oral conscious sedation to deep intravenous (IV) sedation or general anesthesia, depending on institutional protocols, resources, and patient characteristics [4,5,6]. Deep IV sedation can be performed in dedicated non-operating room anesthesia (NORA) settings, which may improve procedural conditions, while oral conscious sedation represents a less invasive alternative. However, evidence comparing these strategies in the specific setting of pediatric kidney transplant biopsies is scarce, and no consensus currently exists on the optimal approach.

This study addresses this gap by comparing oral conscious sedation and deep IV sedation performed in NORA in a large cohort of pediatric kidney transplant recipients undergoing percutaneous biopsy. We specifically evaluated their impact on diagnostic adequacy and complication rates, and we performed a cost-effectiveness analysis to identify the most appropriate strategy for transplanted children and adolescents.

## 2. Materials and Methods

### 2.1. Patients and Data Collection

This retrospective study involved a cohort of kidney recipients aged 0–21 years who underwent kidney biopsy between January 2009 and February 2024 in a single Pediatric Transplant Center. All clinical data were obtained from the medical records, and all patients’ legal guardians gave their written consent for the collection of these data. The Local Ethics Committee authorized the retrospective review of clinical records and data collection for the purposes of the study.

The primary outcome of the study was the diagnostic adequacy of kidney transplant biopsies, defined as the achievement of a final histological diagnosis according to Banff criteria, performed under deep IV sedation in a pediatric NORA setting, and those performed under oral conscious sedation with midazolam.

Secondary outcomes were biopsy-related complications, sedation-related complications, and the cost-effectiveness of the procedure in the same groups.

In renal transplant recipients, biopsies may be performed as a protocol biopsy (i.e., sampling of allograft tissue at specific, predetermined times regardless of function) or as a for-cause biopsy (due to clinical indications such as increased creatinine or pathologic proteinuria). According to our practice, patients undergoing ultrasound-guided percutaneous biopsy are admitted to the Pediatric Nephrology Unit for pre-procedure tests. Blood tests, including complete blood count, prothrombin time (PT), partial thromboplastin time (PTT), and international normalized ratio (INR), are scheduled 24 h prior to the procedure, as well as pediatric anesthesiologic evaluation in case of IV sedation. A pre-procedure renal ultrasound (US) with Doppler evaluation is performed to identify possible contraindications, such as severe hydronephrosis, pre-existing arteriovenous fistula, or cysts located at the lower pole of the allograft. Percutaneous biopsy is performed only if platelet count is >50,000/uL, INR <1.5, and urea <35 mmol/L. Additionally, antiplatelet (e.g., aspirin) and anticoagulant therapy have to be suspended prior to the procedure. Lidocaine and prilocaine topical anesthetic cream are applied to the targeted area one hour before the biopsy, and a single IV dose of cefazolin is administered as antibiotic prophylaxis immediately before the procedure. Biopsies are performed under US guidance using an automated 16-gauge needle. Deep IV or oral conscious sedation is administered according to the patient’s age and compliance or to the family’s preference [7]. All patients included in the study were classified as ASA I–III. The choice between deep IV sedation and oral conscious sedation followed our institutional protocol and was based mainly on patient age, expected compliance, and, in some cases, family preference. In general, younger children (<15 years) and those with anticipated limited cooperation or specific anatomical features that could make the procedure more challenging (e.g., thick abdominal adipose tissue, weak abdominal wall muscles, spinal deformities) were assigned to deep IV sedation, whereas older, cooperative adolescents were considered for oral conscious sedation, even though the final decision rested with the medical team, taking into account patient safety and the likelihood of obtaining an adequate sample. Procedures under intravenous sedation were performed in the NORA setting of our tertiary pediatric hospital. The service is managed by pediatric intensivists and nurses trained in sedation monitoring, with an anesthesiologist immediately available on-site. The NORA performs sedations in accordance with national and international pediatric sedation guidelines, in particular according to our Center’s protocol. IV sedation was induced by midazolam (dose 0.05–0.1 mg/kg, max 5 mg), ketamine (dose 0.5–2 mg/kg), and propofol (dose 0.5–3 mg/kg). Oral conscious sedation was induced by injectable midazolam solution administered orally, diluted in a small volume of sweetened liquid to improve palatability (dose 0.5 mg/kg, max 15 mg). Topical analgesia with lidocaine and prilocaine cream was applied one hour before the procedure in both groups. Local anesthetic (agent, 1% lidocaine) was infiltrated at the biopsy site prior to the procedure for patients under IV sedation. For patients sedated in the NORA setting after 2018, sedation depth and patient cooperation were assessed using the Pediatric Sedation State Scale (PSSS), recorded at baseline, during the procedure, and at 15 min intervals post-procedure [8]. IV sedative drug doses were carefully titrated to achieve a PSSS score of 2 (quiet, not moving, no signs of pain/anxiety) to 3 (expression of pain/anxiety on face, but not moving/impeding completion), which corresponds to a level of sedation that falls within the definition of deep sedation, not general anesthesia. The goal was to maintain patient cooperation and immobility while preserving spontaneous respiration. PSSS data were not available for patients undergoing oral conscious sedation performed by nephrologists. Therefore, PSSS was not included in the comparative analysis due to incomplete availability across the entire study population.

No patients in the NORA group required physical restraint during the procedure. Documentation regarding physical restraint was not consistently available for patients managed under oral sedation, and this limitation has been acknowledged.

Two cortical tissue samples are collected as per guidelines, unless deemed risky due to the patient’s conditions (e.g., proximity to large vessels or hematoma formation) [9]. Reasons for limiting the biopsy are documented in the medical records. After the procedure, the patient undergoes a 24 h monitoring in the Pediatric Nephrology Unit, and a renal ultrasound with Doppler evaluation is performed before discharge.

Data collected for the analysis included patients’ age, gender, biopsy indication, technical difficulties during sample collection, total length of the cortical portion of the specimen, number of glomeruli, histological diagnosis (according to the 2018 Banff score) [9]. According to the 2018 Banff Classification for Renal Allograft Pathology, a biopsy was considered adequate if it contained at least 6 glomeruli and 1 artery; specimens with fewer glomeruli were classified as suboptimal, and those lacking diagnostic material were deemed inadequate. Histological findings were assigned to Banff categories 1–6, which reflect distinct types and severity of allograft pathology and inform subsequent clinical management [9]. Sedation-related complications were classified according to the Tracking and Reporting Outcomes of Procedural Sedation (TROOPS) tool [10], which categorizes events as mild (minor, requiring minimal intervention), moderate (intermediate, requiring active management but no escalation of care), or severe (sentinel, life-threatening or requiring advanced airway or hemodynamic support).

### 2.2. Statistical Analysis

Categorical variables were summarized using frequency distributions and proportions. Numerical variables were described using means (±standard deviation) and medians (interquartile range [IQR]), depending on their distribution. Hypotheses concerning continuous variables were tested using nonparametric methods, such as the Wilcoxon–Mann–Whitney test. The Pearson correlation test was applied to explore potential relationships between variables. All tests were conducted at a 5% significance level, with two-sided confidence intervals calculated at the 95% confidence level. To further highlight the factors influencing diagnostic outcomes in pediatric kidney biopsies, a multiple linear regression analysis was conducted with diagnosis as the dependent variable. The independent variables included in the model were age, indication for biopsy, difficulty during sample collection, biopsy complications, and the type and the degree of sedation. Statistical analysis was performed with the IBM^®^ SPSS^®^ 25 Statistics program, Version 25.0 (IBM Corp., Armonk, NY, USA). Cost-effectiveness analysis was performed using the incremental cost-effectiveness ratio (ICER), calculated as the difference in costs assessed according to the diagnosis-related groups (DRGs) of Italian National Health Care System for the two procedures, assuming a two-day hospital stay (except in case of complications), divided by the difference in effectiveness (intended as successful diagnosis rate) for the two groups (IV sedation and oral sedation) [11].

## 3. Results

### 3.1. Population Characteristics

A total of 711 biopsies performed on 251 pediatric kidney transplant recipients between January 2009 and February 2024 were included in the analysis (Table 1). The mean age of the patients was 12 ± 5.8 years. According to the indication, most biopsies were for protocol (68.4%), while 31.6% were for-cause biopsies. Technical difficulties during sampling were encountered in 150 cases (21.1%), mainly due to patient-specific characteristics of the patient (e.g., conspicuous abdominal adipose tissue, poor abdominal muscles, or spinal dysmorphisms with deformity). The biopsy was performed under deep IV sedation in 548 cases (77.1%), while it was performed under oral conscious sedation in the remaining 163 cases (22.9%). The median length of the cortical part of the specimen was 1.6 cm (IQR 1.2–2.0 cm), with a median number of 13 glomeruli per sample (IQR 7–20).

Regarding diagnoses, 41.5% of the biopsies were classified as Banff 1, followed by Banff 2 (8.3%), Banff 3 (7.3%), Banff 4 (10.4%), Banff 5 (9.1%), Banff 6 (4.2%), and other diagnoses (4.6%). In 14.6% of cases, the sample was deemed inadequate (Table 2).

Biopsy-related complications occurred in 12.9% of procedures (n = 92), and 1.4% experienced more than one complication (Figure 1). Among complications, 50% (6.7% of total biopsies) were isolated gross hematuria, 28.4% (3.7% of total biopsies) by an arteriovenous fistula (2 out of 26 required endovascular embolization), and 20.6% (2.7% of total biopsies) by hematoma (with significant anemia requiring red blood cell transfusion in 1 case). No patients experienced graft loss or death. In summary, the major complications accounted for 4% of total complications, and they included arteriovenous fistula requiring endovascular embolization (0.3% of total biopsies), blood transfusion due to massive hematoma (0.15% of total biopsies), and acute kidney injury (AKI) (0.15% of total biopsies).

Sedation-related complications were reported in 4.9% of cases within the deep IV sedation group. Of these, 44.4% (n = 12) were classified as mild according to TROOPS, including mild desaturation, induction apnea, and mild to moderate hypertension, while 55.6% (n = 15) were classified as moderate, including apnea with desaturation requiring bag-valve-mask ventilation, acute laryngospasm, and acute bronchospasm. None of the patients presented severe complications requiring advanced airway management or admission to the Pediatric Intensive Care Unit (PICU). Five patients (0.7%) experienced recovery agitation after sedation, which resolved spontaneously. In the oral conscious sedation group, one patient experienced a mild desaturation.

### 3.2. Comparison Between the IV Sedation Group and the Oral Conscious Sedation Group

The mean age of the children in the IV sedation group was significantly lower than that of the oral conscious sedation group (10.1 ± 5.0 years vs. 18 ± 2.3 years, *p* < 0.001), indicating that deep IV sedation was more frequently used in younger patients. Regarding the indication for biopsy, there was no significant difference between the groups: for-cause biopsies accounted for 31.4% of biopsies in the IV sedation group compared to 32.5% in the oral conscious sedation group (*p* = 0.91). Similarly, the proportion of protocol biopsies was comparable between the two groups (68.6% in the IV sedation group vs. 67.5% in the oral conscious sedation group, *p* = 0.90). Technical difficulties during sampling were significantly more common in the IV sedation group, affecting 23.2% of biopsies compared to 14.1% in the oral conscious sedation group (*p* < 0.001, OR 1.83 [1.13, 2.97]). The cortical part of the specimen was significantly longer in the IV sedation group, with a median length of 1.8 cm (IQR 1.2–2.0) compared to 1.5 cm (IQR 1.0–2.0) in the oral conscious sedation group (*p* < 0.001). Accordingly, the number of glomeruli was significantly higher in the IV sedation group, with a median of 16 glomeruli (IQR 8–16) compared to 8 glomeruli (IQR 6–15) in the oral conscious sedation group (*p* < 0.001) (Table 2).

The rate of diagnosis yield was significantly higher in the IV sedation group (88.5% of adequate samples compared to 75% in the oral conscious sedation group, *p* < 0.001, OR 2.60 [1.66, 3.97]) (Figure 2).

Biopsy-related complications were similar between the two groups (13.3% in the IV sedation group and 11.6% in the oral conscious sedation group, *p* = 0.58) (Table 3 and Table 4). As expected, sedation-related complications were higher in the IV sedation group, occurring in 4.9% of cases compared to 0.1% in the oral conscious sedation group (*p* = 0.01, OR 8.78 [1.19, 65.04]) (Table 3 and Table 4). In the subgroup of 258 patients aged ≥15 years, the comparison between IV and oral sedation confirmed the findings observed in the overall cohort, with a higher specimen adequacy (89.8% vs. 70.6%, *p* < 0.01) and greater number of glomeruli (12 vs. 8, *p* < 0.01) in the IV sedation group, despite a similar rate of complications and a slightly higher frequency of technical difficulties.

### 3.3. Multiple Linear Regression

The regression model explained 9.7% of the variance in diagnosis classification (R^2^ = 0.097, adjusted R^2^ = 0.090), with a standard error of 0.340. The model was statistically significant (F(6, 704) = 12.637, *p* < 0.001), indicating that the set of predictors accounted for a significant amount of the variation in the diagnosis (Table 5).

Multiple linear regression highlighted that deep IV sedation (β = 0.152, *p* < 0.001), technical difficulties (β = −0.087, *p* = 0.017) and length of the cortical part the specimen (β = 0.208, *p* < 0.001) significantly influenced the likelihood to achieve the final histological diagnosis: longer cortical portion of the specimen and deep IV sedation positively correlated with higher diagnosis achievement scores, while technical difficulties were negatively associated. Age (β = −0.020, *p* = 0.663), indications for the biopsy (β = −0.054, *p* = 0.132), and biopsy-related complications (β = −0.057, *p* = 0.114) did not significantly impact the achievement of a diagnosis.

### 3.4. Anesthesiologic Factors and Their Association with Diagnostic Success, Biopsy-Related Complications, and Sedation-Related Complications in the IV Sedation Group

In the subgroup of 146 patients undergoing IV sedation with complete anesthesiologic data available, the median doses administered were 0.1 mg/kg (IQR 0.08–0.1) for midazolam, 1 mg/kg (IQR 0.8–1.1) for ketamine, and 1.6 mg/kg (IQR 1.08–2.4) for propofol. The median Procedural Sedation State Scale (PSSS) score was 2 (IQR 2–3), indicating an optimal sedation level, and the median procedural duration was 18 min (IQR 15–20).

Regarding biopsy-related complications, no significant association was observed between the administered doses of midazolam (*p* = 0.779), ketamine (*p* = 0.567), or propofol (*p* = 0.663), nor with the procedure duration (*p* = 0.399) or in PSSS (*p* = 0.451). Similarly, for sedation-related complications, no significant differences emerged with respect to midazolam (*p* = 0.538), ketamine (*p* = 0.176), or propofol doses (*p* = 0.312), the duration of the procedure (*p* = 0.416), or in PSSS (*p* = 0.815). In the analysis of diagnostic success, a significantly higher propofol dose was observed in biopsies yielding a histological diagnosis compared to non-diagnostic biopsies (median 1.73 mg/kg vs. 0.95 mg/kg, *p* = 0.013). No significant associations were found for midazolam dose (*p* = 0.288), ketamine dose (*p* = 0.985), procedure duration (*p* = 0.959), or PSSS score (*p* = 0.911).

### 3.5. Cost-Effectiveness Analysis

The costs of the procedure in our setting were found to be the same in both the IV sedation and the oral conscious sedation group, amounting to EUR 1100 per admission for kidney biopsy. The use of IV sedation and anesthesiologic assistance did not affect the cost or the length of hospital stay. Additionally, the diagnostic success rate was higher in the IV sedated group as previously reported (88.5% vs. 75%, *p* < 0.001).

The cost-effectiveness analysis showed a cost per successful diagnosis of 1243 EUR in the deep IV sedation group and 1467 EUR in the oral conscious sedation group. Since both procedures have the same cost, the ICER was zero, indicating that the procedure under deep IV sedation provides higher diagnostic success without additional cost.

## 4. Discussion

Our study is the first to compare the use of oral conscious sedation and deep IV sedation in NORA on a large number of allograft biopsies performed in children and adolescents.

The key finding is a 2.6-fold higher success rate in patients who underwent biopsy under deep IV sedation in NORA compared to those who received oral conscious sedation. The overall diagnostic success rates of ultrasound-guided percutaneous kidney biopsies, including both transplanted and native kidneys, have been reported to range between 87.2% and 100% [4,12,13,14,15]. In our study, the diagnostic rate was notably lower (75%) in children who received oral conscious sedation compared to the deep IV sedation group (88.5%). Shin et al. analyzed 448 percutaneous renal allograft biopsies in adult recipients who received only local lidocaine, reporting a diagnostic rate of 84.6%, higher than that observed in our study in the oral conscious sedation group (who received oral midazolam and local lidocaine) [13]. This difference may be explained by greater cooperation in adults compared to children or adolescents. Aside from this study, the few studies available in the literature present significant limitations in terms of comparability of results, particularly due to the inclusion of native kidney biopsies, the use of general anesthesia during the procedure, and the involvement of different professional groups, such as radiologists or pediatric nephrologists or not specified, in performing the biopsies across studies [4,12,13,14,15]. Thus, our study is the first one to report the diagnostic success rate in children who received oral conscious sedation and deep IV sedation.

Children who underwent the procedure under deep IV sedation, with longer cortical biopsy samples and therefore a greater number of glomeruli, had a higher diagnostic rate, and the length of the cortical part of the sample and the number of glomeruli directly correlated with the diagnostic success rate. This result is explained by a minimum requirement of 6 glomeruli for a kidney transplant biopsy to be considered diagnostic according to Banff criteria [9]. Consequently, technical difficulties during sample collection negatively affect the procedure, compromising the quality of the sample, as emerged in our analysis.

As expected, age was significantly lower in patients who underwent deep IV sedation due to lower procedural compliance among younger children. In the multivariate analysis, age was not independently associated with diagnostic yield; however, given the strong correlation between age and sedation type, the independent contribution of age could not be fully disentangled. In the subgroup analysis of patients aged ≥15 years, the comparison between IV and oral sedation confirmed the results observed in the overall cohort, with higher specimen adequacy and diagnostic success in the IV sedation group. The slight but statistically significant age difference between groups was not clinically relevant, as in our clinical practice, patients aged 15 years and older are routinely offered both sedation options, with the final choice based on patient preference and expected compliance.

Furthermore, in the subgroup analysis of patients undergoing IV sedation, a higher administered dose of propofol was the only anesthesiologic factor significantly associated with an increased diagnostic success rate in the IV sedation group. This suggests that achieving an optimal depth of sedation, sufficient to ensure patient immobility without oversedation, may play a crucial role in maximizing biopsy performance and histological adequacy. The higher rate of technical difficulties observed in the IV sedation group likely reflects patient-related factors rather than the sedation method itself. In our setting, younger children and patients with specific anatomical characteristics that make percutaneous biopsy more challenging (e.g., thick abdominal adipose tissue, weak abdominal wall muscles, or skeletal deformities) are preferentially assigned to deep IV sedation. In line with our subgroup analyses, these technical difficulties appear to be mainly associated with the younger age of patients undergoing IV sedation, rather than with the sedation technique per se, which otherwise allowed for better procedural control and higher specimen adequacy.

Complications of percutaneous kidney biopsy can vary from minor perirenal hematomas or gross hematuria to severe bleeding that requires blood transfusions or interventional radiology procedures and may even result in kidney loss. Patient deaths have been reported in rare cases [6]. The overall complication rate in our population was 12.9%, which aligns with the literature, reporting a range of 7–17% [16]. However, the incidence of severe complications, defined as the need for surgical intervention or blood transfusion, was lower in our population compared to other studies (0.3% and 0.15% in our study, versus 0.4–1.1% and 0.5–2.0%, respectively) [16,17]. However, these earlier studies also included biopsies on native kidneys, and specific data on complications in biopsies performed on kidney grafts are lacking [12,13,14,15,16,17]. Nevertheless, a recent meta-analysis on the complications of renal biopsies in transplanted kidneys demonstrated an overall complication rate lower than in our study (6.0% vs. 12.9%), as well as a lower incidence of minor complications, such as macrohematuria (3.18% vs. 6.5% in our population), arteriovenous fistula (1.5% vs. 3.7%), and hematoma (1.6% vs. 2.7%) [18]. However, this meta-analysis primarily involved adults, while data reported in a subgroup of children were comparable to our study’s results, highlighting a lower incidence of major complications and the need for blood transfusions in pediatric patients and emphasizing the overall safety of percutaneous allograft biopsy in children.

The cost-effectiveness analysis indicated that deep IV sedation is more effective, primarily because the costs of performing the biopsy under deep IV sedation and oral conscious sedation are the same. In fact, renal biopsy requires hospital admission, regardless of sedation, because 24 h monitoring for potential complications is recommended [19,20]. Hospital admission itself accounts for the main economic factor, but it represents a fixed fee regardless of the sedation method. Furthermore, as no patients required PICU admission or prolonged hospitalization due to sedation-related complications, no impact on the costs of hospitalization was observed.

Furthermore, while deep IV sedation seems to have a higher diagnostic yield without increasing overall costs, it may also enhance patient comfort during the procedure and reduce anxiety and distress [21]. However, previous research mainly focused on complications or differences between distinct techniques and settings, and did not specifically examine the impact of different sedation strategies [4,14,22,23,24,25]. Many studies report data on renal biopsies performed under general anesthesia, and deep IV sedation, even if used, has not been thoroughly studied as a possible contributor to improved procedural outcomes [4,14,17,18,19]. A recent paper compared two different approaches in inducing general anesthesia for renal biopsy, showing that sedation–aspiration combined anesthesia led to fewer hemodynamic fluctuations and improved postoperative recovery in children compared to total intravenous anesthesia [20]. In our cohort, patients who underwent deep IV sedation only presented mild to moderate sedation-related adverse events without requiring advanced management of airways or hemodynamic support. The possibility of performing biopsies in a designated secure procedural sedation setting outside of the operating room may improve children’s overall well-being by enhancing patient comfort, boosting diagnostic accuracy, and providing a cost-effective alternative.

This study has several limitations. First, its retrospective design and single-center setting may limit the generalizability of the findings. However, the main limitation of our study is the lack of a uniform assessment of sedation depth across the entire cohort. Although from 2018 onwards the PSSS was systematically recorded for patients undergoing kidney biopsy under IV sedation in the NORA setting, this information was not available for procedures performed before 2018 or for those conducted under oral conscious sedation managed by nephrologists [8]. Consequently, sedation depth and cooperation could not be uniformly assessed across the entire study cohort [26]. Although uniform assessment of sedation depth was not possible, available data from the NORA subgroup confirmed that sedation was delivered within the intended range of deep but not general anesthesia. However, the present study was not designed to investigate sedation depth as a primary endpoint, but rather to evaluate how different sedation modalities can influence biopsy adequacy, safety, and cost-effectiveness. Future prospective studies are warranted to include standardized and comparable measures of sedation depth and patient comfort across modalities.

In addition, systematic evaluation of pre- and post-procedural anxiety using validated psychometric tools was not available for the retrospective period analyzed. Although in recent years validated scales such as the State-Trait Anxiety Inventory for Children, Wong–Baker Faces Pain Scale, and Children’s Fear Scale have been incorporated into our NORA routine practice in collaboration with clinical psychologists, these tools were not in use during most of the study period and therefore could not be included in the analysis [27,28,29]. In addition, the choice between deep IV sedation and oral conscious sedation was based on clinical judgment, patient characteristics, and family preference rather than random allocation. Therefore, unmeasured confounding factors may have influenced sedation group assignment and, consequently, biopsy outcomes, introducing potential selection bias. Moreover, data regarding the need for physical restraint during the procedure were not consistently documented outside the NORA setting, limiting the ability to fully assess procedural comfort and patient experience. These limitations highlight the need for future prospective studies integrating systematic assessment of sedation quality, anxiety levels, and patient-reported outcomes to optimize procedural care and to validate the observed associations. These considerations further underscore that the retrospective design inherently limits the completeness and reliability of sedation-related data, which should be taken into account when interpreting the study findings.

Finally, the ability to provide structured, effective sedation for children undergoing percutaneous procedures should not be viewed as an ancillary support, but as a fundamental prerequisite for achieving optimal diagnostic outcomes and ensuring patient safety and comfort.

## 5. Conclusions

Our study suggests that percutaneous allograft biopsy under deep intravenous sedation improves diagnostic success in pediatric kidney-transplant recipients without increasing costs or complication rates. Among anesthetic factors, a higher propofol dose was associated with better diagnostic yield, highlighting the importance of achieving adequate sedation depth. These results must be interpreted in light of the retrospective design and incomplete data on sedation quality and patient experience. Finally, our findings highlight that the availability of structured pediatric sedation services may play an important role in improving the quality and diagnostic success of kidney biopsies in children with kidney transplants.

## Figures and Tables

**Figure 1 jcm-14-07361-f001:**
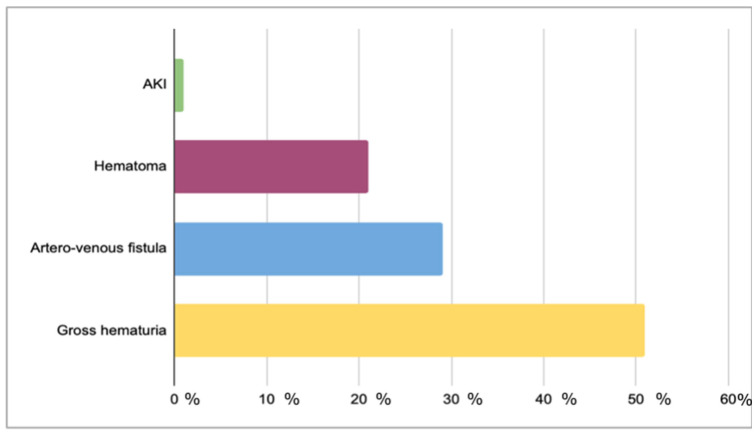
Types of renal biopsy complications and their respective percentage weight of total complications. AKI: acute kidney injury.

**Figure 2 jcm-14-07361-f002:**
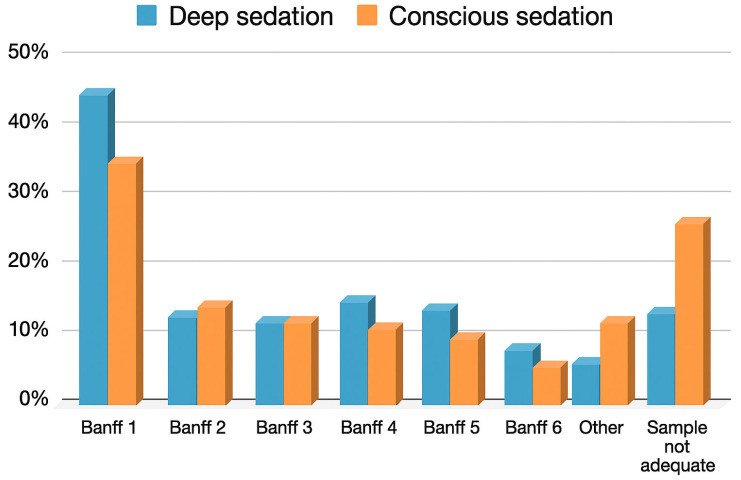
Distribution of histological diagnoses according to the type of sedation.

**Table 1 jcm-14-07361-t001:** Characteristics of the patients.

Patients	n = 251
Biopsies	n = 711
Mean age of patients	12 ± 5.8 years
Sex	Male 143 (57%), Female 108 (43%)
For-cause biopsy	n = 225 (31.6%)
Protocol biopsy	n = 486 (68.4%)
Technical difficulties during sampling	n = 150 (21.1%)
Length of the cortical part of the specimen	1.6 cm (IQR 1.2–2.0)
Median number of glomeruli	13 (IQR 7–20)

**Table 2 jcm-14-07361-t002:** Characteristics of the patients according to the type of sedation.

n = 711	IV Sedation	Oral Sedation	*p*-Value
Biopsies	n = 548	n = 163	
Mean age	10.1 ± 5.0 years	18 ± 2.3 years	<0.001
For-cause biopsy	n = 172 (31.4%)	n = 53 (32.5%)	0.91
Protocol biopsy	n = 376 (68.6%)	n = 110 (67.5%)	0.90
Technical difficulties in sampling	n = 127 (23.2%)	n = 23 (14.1%)	<0.001
Length of the cortical part of the specimen	1.8 cm (IQR 1.2–2.0)	1.5 cm (IQR 1.0–2.0)	<0.001
Median number of glomeruli	16 (IQR 8–16)	8 (IQR 6–15)	<0.001

**Table 3 jcm-14-07361-t003:** Complications according to the type of sedation.

	IV Sedation	Oral Sedation	*p*-Value
Biopsy-related complications	n = 73 (13.3%)	n = 19 (11.6%)	0.58
Sedation-related complications	n = 27 (4.9%)	n = 1 (0.1%)	0.01

**Table 4 jcm-14-07361-t004:** Characteristics of the patients according to the type of sedation in patients aged > 15 years old.

	IV Sedation	Oral Sedation	*p* Value
	(N = 108)	(N = 150)	
Age	16.8 years (IQR 15.9–18.0)	18.8 years (IQR 17.2–20.4)	*p* = 0.013
Protocol biopsy	64/108 (59%)	102/150 (68%)	*p* = 0.152
Technical difficulties	26/108 (24%)	22/150 (15%)	*p* = 0.062
Length of the cortical part of the specimen	2.0 cm (IQR 1.4–2.1)	1.5 cm (IQR 1–2)	*p* < 0.01
Median number of glomeruli	12. (IQR 7–19)	8.0 (IQR 5–14)	*p* < 0.01
Biopsy-related complications	14/108 (13%)	0.1 19/150 (13%)	*p* = 0.942
Diagnostic biopsy	97/108 (89.8%)	106/150 (70.6%)	*p* < 0.01

**Table 5 jcm-14-07361-t005:** Multiple linear regression.

Variable	Beta (β)	*p*-Value	Significance
Age	−0.020	0.663	Not significant
For cause-biopsy	−0.054	0.132	Not significant
Technical difficulties	−0.087	0.017	Significant, negative relationship
Length of the cortical part of the specimen	0.208	<0.001	Significant, positive relationship
Biopsy-related complications	−0.057	0.140	Not significant
Sedation	0.152	<0.001	Significant, positive relationship

## Data Availability

The datasets are not publicly available due to privacy and ethical restrictions. However, they are available from the corresponding author on reasonable request and with permission from the Ethics Committee, if applicable.

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
