# Peer review of "Percutaneous Biopsy Under Deep Intravenous or Oral Conscious Sedation: Which Is the Best Option for Pediatric Renal Transplant Recipients?"

_jcm, 2025, doi:10.3390/jcm14207361_

Round 1
Reviewer 1 Report
Comments and Suggestions for Authors
This paper provides a comprehensive and timely article titled on “Percutaneous biopsy under deep intravenous or oral conscious sedation: which is the best option for pediatric renal transplant recipients”. I recommend minor revision. The abstract could be more concise. The introduction lacks clarity and focus; it should better highlight the clinical importance of pediatric kidney transplant biopsies, the role of sedation, and the specific gap this study addresses to provide a stronger rationale for the research. Authors need to provide a more detailed description of the sedation protocols, including propofol dosing and methods used to monitor sedation depth during the biopsy procedures. Does the retrospective study design influence the reliability and completeness of the data, particularly in relation to sedation quality and patient outcomes. Authors need to provide more details on how sedation quality was assessed or documented, and how missing or incomplete sedation data may impact the interpretation of the findings. Consider analyzing whether patient-specific factors like age, weight, or comorbidities influenced sedation requirements or biopsy outcomes.
Author Response
We thank the reviewer for the valuable suggestions. Here we provide a point-by-point response to the reviewer’s comments.
Comment 1:
The abstract could be more concise.
Answer 1:
We thank the reviewer for this helpful suggestion. We have revised the abstract to make it more concise while retaining all key results and conclusions (see Abstract, page 1, highlighted in green).
Comment 2:
The introduction lacks clarity and focus; it should better highlight the clinical importance of pediatric kidney transplant biopsies, the role of sedation, and the specific gap this study addresses to provide a stronger rationale for the research.
Answer 2:
We thank the Reviewer for this valuable comment. We have thoroughly revised the Introduction to emphasize the clinical importance of kidney allograft biopsy in pediatric patients, the critical role of sedation in ensuring both diagnostic adequacy and patient safety, and the lack of comparative evidence regarding different sedation strategies in this specific population. The revised Introduction now clearly outlines the rationale for the present study and the knowledge gap it addresses (see Introduction, page 2).
Comment 3:
Authors need to provide a more detailed description of the sedation protocols, including propofol dosing and methods used to monitor sedation depth during the biopsy procedures.
Answer 3:
We thank the reviewer for this valuable comment. All details regarding the sedation protocol, including drug combinations, dosing, and monitoring methods, are now clearly described in the Methods section. Briefly, deep IV sedation was performed with a combination of midazolam, ketamine, and propofol, titrated to achieve a Pediatric Sedation State Scale (PSSS) level of 2–3, corresponding to deep sedation while maintaining spontaneous respiration. For procedures performed after 2018, sedation depth was systematically monitored using the PSSS at baseline, during the procedure, and post-procedure. This information has been clarified and expanded in the revised Methods section. (see page 4, lines 149-153, highlighted in green).
Comment 4:
Does the retrospective study design influence the reliability and completeness of the data, particularly in relation to sedation quality and patient outcomes. Authors need to provide more details on how sedation quality was assessed or documented, and how missing or incomplete sedation data may impact the interpretation of the findings. Consider analyzing whether patient-specific factors like age, weight, or comorbidities influenced sedation requirements or biopsy outcomes.
Answer 4:
We thank the reviewer for this comment. As stated in the manuscript (see Discussion section), the retrospective design limits the completeness and reliability of sedation-related data. Systematic PSSS recording was only available from 2018 onwards for IV sedation in the NORA setting, while data for oral conscious sedation or procedures performed before 2018 were not consistently documented. Consequently, sedation depth, patient cooperation, and procedural comfort could not be uniformly assessed across the entire cohort, which may influence the interpretation of the findings. We have now explicitly highlighted these limitations (Discussion, page 12, lines 435-439).
Regarding patient-specific factors, we analysed the potential influence of age, biopsy indication, and technical difficulties on diagnostic yield through multivariate linear regression. Furthermore, we add a subgroup analysis for patients aged > 15 years when both kinds of sedation are considered in clinical practice. In this comparison between IV and oral sedation confirmed the results observed in the overall cohort, with higher specimen adequacy and diagnostic success in the IV sedation group. (Results, page 7, lines 267-271 and table 4).
Reviewer 2 Report
Comments and Suggestions for Authors
Thanks for the opportunity to review this manuscript. I think that this is valuable. Obviously there are limitations in terms of the retrospective design, but overall I think that the conclusions seem mostly sound. I have some concerns about the interpretations however, see below for more details. I am in particular questioning whether more thought needs to be given to the decision making to provide IV rather than oral. If this is the clinician's choice, then there are existing biases and factors that would determine sedation allocation. These might not necessarily be all listed in this manuscript, meaning that the group allocation - and therefore possibly also the conclusions - are not free from bias.
The Banff criteria/cut offs and implications should be described in the methods section with an appropriate citation
Sedation-related complications: Can it be confirmed that only one occurred in the oral group?
Figure 2 should be clearer in terms of that it is for the overall sample. I wonder whether there's a way to convey how many were from the IV group
There was age-related decision making in terms of sedation type. This is a possible confound (though the authors rule this out). However, are there other potential reasons for the decision to give IV or oral sedation outside of the data here that could account for the differences in factors such as technical difficulty in sampling, length of cortical part, or number of glomeruli? E.g., health or condition related factors? There could be confounding factors underlying diagnosis success that were also associated with decision making for sedation type
The different drugs may also have affected success rather than the route of administration
Why do the authors think that higher technical difficulties occurred during IV sedation, when there should be less interference from the patient?
" However, age itself did not impact the diagnostic yield of the biopsy" - this may not be accurate. What is the range of the patients receiving oral? If there were no (or few) younger patients receiving oral conscious sedation, and all patients in the IV condition were fully sedated, then age cannot be accurately measured since most patients should display similar behavior in the IV condition
Author Response
We thank the reviewer for the valuable suggestions. Here we provide a point-by-point response to the reviewer’s comments.
Comment 1:
I am in particular questioning whether more thought needs to be given to the decision making to provide IV rather than oral. If this is the clinician's choice, then there are existing biases and factors that would determine sedation allocation. These might not necessarily be all listed in this manuscript, meaning that the group allocation - and therefore possibly also the conclusions - are not free from bias.
Answer 1:
We thank the reviewer for this thoughtful comment. We agree that the allocation to deep IV sedation versus oral conscious sedation was not random but based on pragmatic clinical criteria. To clarify this point, we have expanded the Methods section to provide a more detailed description of our institutional protocol for sedation assignment, which primarily relies on patient age, expected compliance, and, in some cases, family preference (see Materials and Methods, Section 2.1 “Patients and data collection”, page 3, lines 127-134, highlighted in green). We acknowledge that this assignment method is not random and introduces a potential for selection bias, which we addressed in our statistical analysis by comparing patient characteristics between the two groups (Table 2) and by including relevant confounding factors in our multivariate analysis. In addition, we have explicitly mentioned this potential bias in the Discussion, under the study limitations paragraph (see page 12, lines 435-439, highlighted in green).
Comment 2:
The Banff criteria/cut offs and implications should be described in the methods section with an appropriate citation.
Answer 2:
We thank the reviewer for this suggestion. We have revised the Methods section to provide a clearer description of the Banff criteria used to define biopsy adequacy and classify histological findings, including the relevant cut-offs and their clinical implications (see Materials and Methods, Section 2.1 “Patients and data collection”, page 4, lines 170-175, highlighted in green).
Comment 3:
Sedation-related complications: Can it be confirmed that only one occurred in the oral group?
Answer 3:
We confirm that only one sedation-related complication occurred in the oral conscious sedation group and consisted of a mild, self-limiting desaturation episode.
Comment 4:
Figure 2 should be clearer in terms of that it is for the overall sample. I wonder whether there's a way to convey how many were from the IV group.
Answer 4:
We thank the reviewer for this valuable comment. Considering that Figure 2 was not sufficiently clear and that the number of individual complications was very low, we decided to remove the figure and report the complications directly in the Results section for greater clarity and readability (see Results Section 3.1, “Population characteristics”, page 5, lines 227-230, highlighted in green).
Comment 5:
There was age-related decision making in terms of sedation type. This is a possible confound (though the authors rule this out). However, are there other potential reasons for the decision to give IV or oral sedation outside of the data here that could account for the differences in factors such as technical difficulty in sampling, length of cortical part, or number of glomeruli? E.g., health or condition related factors? There could be confounding factors underlying diagnosis success that were also associated with decision making for sedation type. The different drugs may also have affected success rather than the route of administration
Answer 5:
We agree that age and patient characteristics may have influenced the choice of sedation type. In our practice, IV sedation is preferentially used in younger or less cooperative patients and in those with anatomical factors increasing procedural complexity, which may explain the higher rate of technical difficulties. However, in the subgroup analysis of patients aged ≥15 years that we added, where both sedation options are routinely proposed, the results were consistent with the overall cohort, showing higher specimen adequacy (2.0 vs. 1.5 cm, p<0.01) and more glomeruli (12 vs. 8, p<0.01) in the IV sedation group. This supports that the improved diagnostic yield mainly reflects the procedural advantages of deeper sedation rather than confounding factors (see page 7, lines 267-271, and table 4 highlighted in green).
Comment 6:
Why do the authors think that higher technical difficulties occurred during IV sedation, when there should be less interference from the patient?
Answer 6:
We thank the reviewer for this insightful comment. The higher rate of technical difficulties in the IV sedation group reflects patient-related factors rather than the sedation method. In our setting, younger children and patients with specific anatomical features that make percutaneous biopsy more challenging are preferentially assigned to deep IV sedation, which explains this finding. We have clarified this point in the Discussion section (see page 11, lines 369-377, highlighted in green).
Comment 7:
However, age itself did not impact the diagnostic yield of the biopsy" - this may not be accurate. What is the range of the patients receiving oral? If there were no (or few) younger patients receiving oral conscious sedation, and all patients in the IV condition were fully sedated, then age cannot be accurately measured since most patients should display similar behavior in the IV condition.
Answer 7:
We agree with the reviewer with this point. We have revised the comment on the multivariate analysis in the Discussion to clarify that, due to the strong correlation between age and sedation type, the independent contribution of age could not be fully disentangled (see Discussion, page 10, lines 355-363, highlighted in green).
Reviewer 3 Report
Comments and Suggestions for Authors
In this manuscript authors did a retrospective comparison of different sedation approaches and its effect on biopsy outcomes in pediatric kidney transplant patients.
My major concern for this study is the lack of sedation measurement. If we are comparing two sedation techniques and their impact on invasive procedures success, then a minimum outcomes measured and compared should be a level of sedation. In this case, there is no data for significant number of patients in iv sedation group and no data for any of the patients in oral sedation group.
Second, authors say that procedures in iv sedation group were more succesful. However, as they mention in Discussion (line 253-254), succesful biopsy is considered as one with at least 6 glomeruli. In both groups analyzed median number of glomeruli was above 6 (16 vs 8) meaning that biopsies in oral sedation group are also considered successful according to the definition.
Regarding the cost of procedures, I presume that iv sedation is performed by anesthesiologist. Therefore, cost of procedure should take this factor into account as oral sedation can be done by nephrologists meaning less staff is needed.
Methods:
Please define what does NORA setting stand for.
Please describe PSSS in more details.
Who performed iv sedation? What kind of monitoring was applied? Please explain in details.
What form of midazolam was used for oral sedation?
Please clearly define primary and secondary outcomes.
In the results section, authors mention "mild, moderate and severe" sedation-related complications. However, these should be clearly defined in the Methods section.
Results
Table 2 could be omitted from the manuscript as this is already described in the text.
Figure 1. - explain abbreviations (AKI). Explain the meaning of the numbers on x-axis (number of patients, percentages...)
How do authors explain increased incidence of technical difficulties in iv sedation group? Also, how do you explain better specimen quality despite more technical difficulties in this group, especially considering linear regression analysis results (lines 192-198).
How did you include depth of sedation in linear regression analysis if there are no data on depth of sedation?
Author Response
We thank the reviewer for the valuable suggestions. Here we provide a point-by-point response to the reviewer’s comments.
Comment 1:
My major concern for this study is the lack of sedation measurement. If we are comparing two sedation techniques and their impact on invasive procedures' success, then a minimum outcome measured and compared should be a level of sedation. In this case, there is no data for a significant number of patients in iv sedation group and no data for any of the patients in the oral sedation group.
Answer 1:
We thank the reviewer for this important comment. Sedation depth and patient cooperation were assessed using the Pediatric Sedation State Scale (PSSS) only for patients sedated in the NORA setting after 2018, when this tool became validated. Most of these patients showed a level 2 of sedation (“quiet, not moving, no signs of pain or anxiety”), while occasional level 3 episodes were promptly corrected by adjusting positioning or drug dosage. However, PSSS data were unavailable for patients sedated orally by nephrologists and for those treated before 2018. For this reason, sedation level was not included in the comparative analysis, as it was inconsistently recorded across the entire cohort (2009–2023). As clarified in the limitations section, validated psychometric tools for assessing procedural anxiety (e.g., STAI-C, Children FS, SPSS) have only been available since 2018. Before that, patient comfort was reported narratively in medical charts, and no cases of significant discomfort or procedure interruption were documented. As far this is the main limitation of the study we underlined in the Discussion (page 12, lines 422-423, highlighted in green).
Comment 2:
Second, authors say that procedures in iv sedation group were more successful. However, as they mention in Discussion (line 253-254), successful biopsy is considered as one with at least 6 glomeruli. In both groups analyzed median number of glomeruli was above 6 (16 vs 8) meaning that biopsies in the oral sedation group are also considered successful according to the definition.
Answer 2:
We thank the Reviewer for this important clarification. We fully agree that according to Banff criteria, a biopsy containing ≥6 glomeruli is considered diagnostic, and indeed 75% of biopsies performed under oral midazolam sedation met this threshold and were deemed adequate. However, our results show that the diagnostic rate was significantly lower in the oral sedation group compared to the IV sedation group (75% vs 88.5%, p < 0.001). This difference highlights that, although the median number of glomeruli was above the diagnostic threshold in both groups, oral sedation was associated with a higher proportion of non-diagnostic biopsies. Furthermore, as we discussed in the manuscript (page 9, lines 268–274), the diagnostic yield observed in the oral sedation group was not only lower than in the IV sedation group, but also inferior to what has been reported in the literature for both pediatric and adult kidney transplant biopsies (87.2-100%).
Comment 3:
Regarding the cost of procedures, I presume that iv sedation is performed by anesthesiologist. Therefore, cost of procedure should take this factor into account as oral sedation can be done by nephrologists meaning less staff is needed.
Answer 3 :
We thank the reviewer for this important observation. In our institution, all NORA procedures are performed by pediatric intensivists without anesthesiologists or operating room staff involvement. Procedure costs were calculated according to the national Diagnosis-Related Group (DRG) system, which provides a fixed reimbursement for kidney biopsy admissions irrespective of the sedation type used. Therefore, within our healthcare model, the cost per procedure was identical for both IV and oral sedation (€1,100), as staff composition does not affect the DRG-based reimbursement. We acknowledge that this reflects our specific national system and may differ from cost structures in other healthcare settings.If the biopsies were performed in the operating room with a full surgical and anesthesiology team, the cost would indeed be higher according to the Italian DRG classification; however, procedures conducted in the Non-Operating Room setting do not entail additional costs under this system. We have now specified in the Methods section that the DRG calculation refers specifically to the Italian national healthcare system (page 4, line 196, highlighted in green). We acknowledge that this reflects our specific national healthcare model and may differ from cost structures in other countries.
Methods:
Comment 4: Please define what does NORA setting stand for. Please describe PSSS in more details. Who performed iv sedation? What kind of monitoring was applied? Please explain in details.
Answer 4:
We thank the reviewer for the opportunity to clarify the characteristics and safety standards of our Non-Operating Room Anesthesia (NORA) service. In our institution, procedural sedation is managed by a dedicated team of pediatric intensivists (average experience >10 years), nurses trained in sedation monitoring, and a pediatric nephrologist, with an anesthesiologist always immediately available on-site for consultation. The service performs over 4,000 procedures annually, including renal and liver biopsies, endoscopies, lumbar punctures, and vascular access, within a tertiary pediatric hospital. All patients with ASA >III are referred to the operating room. Our NORA operates according to national and international guidelines, requiring PALS certification and advanced airway management training for all providers. Over more than 15 years and >60,000 sedations, no severe sedation-related complications have occurred, confirming the safety and robustness of this model. We add a sentence in the Materials and Methods section (see page 3, lines 135-139, highlighted in green). For patients sedated in the NORA setting after 2018 when this scale was validated, sedation depth and patient cooperation were assessed using the Pediatric Sedation State Scale (PSSS), recorded at baseline, during the procedure, and at 15-minute intervals post-procedure. However, PSSS data were not available for patients undergoing oral conscious sedation performed by nephrologists. Therefore, PSSS was not included in the comparative analysis due to incomplete availability across the entire study population. We add a sentence in the Materials and Methods section (see page 4, lines 149-153, highlighted in green).
Comment 5:
What form of midazolam was used for oral sedation?
Answer 5:
According to our practice, oral sedation was performed using the injectable midazolam solution administered orally, diluted in a small volume of sweetened liquid to improve palatability. A commercially available oral formulation was not used, as it is not available in our setting. We have clarified this detail in the revised Methods section (see page 3, lines 141–143).
Comment 6:
Please clearly define primary and secondary outcomes.
Answer 6:
We thank the Reviewer for this helpful remark. We have clarified in the revised manuscript the outcomes and the comparison groups in methods section (see page 3, lines 102-107).
Comment 7:
In the results section, authors mention "mild, moderate and severe" sedation-related complications. However, these should be clearly defined in the Methods section.
Answer 7:
We thank the reviewer for this valuable comment. Sedation-related complications were classified according to the Tracking and Reporting Outcomes of Procedural Sedation (TROOPS) tool (BJA 2018;120:164–172), which standardizes the assessment of adverse events based on their clinical impact and the interventions required. Specifically, mild events are those easily managed with minimal intervention and without risk for the patient (e.g., transient desaturation requiring supplemental oxygen or airway repositioning); moderate events are those that could endanger the patient if not promptly managed or that indicate suboptimal sedation quality (e.g., need for positive pressure ventilation or use of βâ‚‚-agonists); and severe (sentinel) events correspond to life-threatening complications requiring advanced interventions such as tracheal intubation, vasoactive support, or cardiopulmonary resuscitation. We have now specified these definitions in the Methods section (see page 4, lines 175-179). In our cohort, 44.4% of events were mild and 55.6% moderate, with no severe events recorded, consistent with published pediatric sedation safety data.
Results
Comment 8:
Table 2 could be omitted from the manuscript as this is already described in the text.
Answer 8:
We agreed with the suggestion and removed Table 2 from the manuscript.
Comment 9:
Figure 1. - explain abbreviations (AKI). Explain the meaning of the numbers on x-axis (number of patients, percentages...).
Answer 9:
AKI is an abbreviation for acute kidney injury as reported previously in the text and in the glossary. As suggested, the definition was also reported under figure 1. X-axis represents the percentage weight of every single type of kidney biopsy complication over the total complications. We modified the statement of the figure as it is clearer (see Figure 1).
Comment 10:
How do authors explain increased incidence of technical difficulties in iv sedation group? Also, how do you explain better specimen quality despite more technical difficulties in this group, especially considering linear regression analysis results (lines 192-198). How did you include depth of sedation in linear regression analysis if there are no data on depth of sedation?
Answer 10:
The higher rate of technical difficulties observed in the IV sedation group most likely reflects patient-related factors rather than the sedation technique itself. In fact in subgroup analysis (patients aged at least 15 years) no differences were highlighted in terms of technical difficulties. In our clinical practice, deep IV sedation is preferentially used in younger children or in patients with anatomical characteristics that increase procedural complexity—such as increased abdominal adiposity, reduced muscle tone, or skeletal deformities. These factors inherently make percutaneous kidney biopsy more challenging and may account for the higher rate of technical difficulties recorded in this group (see Discussion at page 11, lines 369-377). Nevertheless, IV sedation provided better control of movement and anxiety, allowing operators to obtain longer cortical cores and a higher number of glomeruli, resulting in superior specimen adequacy and diagnostic yield compared with oral sedation despite more technical difficulties. Regarding the regression analysis, depth of sedation was not included as a variable due to incomplete data availability across the entire study period.
Round 2
Reviewer 2 Report
Comments and Suggestions for Authors
No further comments
Author Response
Comment: No further comments.
Reply: We thank the Reviewer.
Reviewer 3 Report
Comments and Suggestions for Authors
In this revised manuscript authors provided their responses to reviewers comments and have made changes accordingly.
However, my major concern remains the lack of measurement of sedation levels. In my opinion, if authors analyze and compare effects and different modalities of sedation, the main outcome/variable should be level of sedation.
Author Response
Comment: In this revised manuscript authors provided their respoReply nses to reviewers comments and have made changes accordingly. However, my major concern remains the lack of measurement of sedation levels. In my opinion, if authors analyze and compare effects and different modalities of sedation, the main outcome/variable should be level of sedation.
Response:
We thank the Reviewer for this suggestion. We agree that sedation depth is a key determinant of procedural quality and patient comfort. However, as clarified in the Introduction and Methods sections, the primary aim of our study was to compare two distinct sedation modalities, deep intravenous versus oral conscious sedation, in terms of diagnostic adequacy, complication rates, and cost-effectiveness, rather than to analyze the level of sedation itself. Because the study was retrospective and spanned a 15-year period, systematic measurement of sedation depth using the Pediatric Sedation State Scale (PSSS) was available only for procedures performed under IV sedation in the NORA setting after 2018, and not for oral sedation. Thus, a direct comparison of sedation levels between the two modalities would have introduced major bias. Nevertheless, for transparency, the available data from the NORA subgroup were reported and analyzed in the Results section, confirming that sedation was delivered within the expected range of deep but not general anesthesia. We have now explicitly emphasized this point in the Discussion and clarified that the study’s focus was on the impact of sedation modality, rather than depth, on procedural and diagnostic outcomes. Future prospective studies including standardized measures of sedation depth and patient comfort will be essential to complement these findings (see (Discussion, page 12, lines 428-435, highlighted in yellow).